# Maternal Folic Acid Intake and Methylation Status of Genes Associated with Ventricular Septal Defects in Children: Case–Control Study

**DOI:** 10.3390/nu13062071

**Published:** 2021-06-17

**Authors:** Sandra M. González-Peña, Geovana Calvo-Anguiano, Laura E. Martínez-de-Villarreal, Patricia R. Ancer-Rodríguez, José J. Lugo-Trampe, Donato Saldivar-Rodríguez, María D. Hernández-Almaguer, Melissa Calzada-Dávila, Ligia S. Guerrero-Orjuela, Luis D. Campos-Acevedo

**Affiliations:** 1Clinical Nutrition, Hospital Universitario “Dr. José Eleuterio González” and Medicine Faculty, Autonomous University of Nuevo León, Monterrey 64460, Mexico; licnut_sandragzz@hotmail.com (S.M.G.-P.); patyaner@hotmail.com (P.R.A.-R.); 2International Iberoamerican University of México, Campeche 24560, Mexico; 3Genetics Department, Hospital Universitario “Dr. José Eleuterio González” and Medicine Faculty, Autonomous University of Nuevo León, Monterrey 64460, Mexico; qfb.geca@gmail.com (G.C.-A.); laelmar@yahoo.com.mx (L.E.M.-d.-V.); lugotramjose@hotmail.com (J.J.L.-T.); dhernandez35@uabc.edu.mx (M.D.H.-A.); calzada_mel@hotmail.com (M.C.-D.); 4Gynecology and Obstetrics Department, Hospital Universitario “Dr. José Eleuterio González” and Medicine Faculty, Autonomous University of Nuevo León, Monterrey 64460, Mexico; donato.saldivar@hotmail.com; 5Medicine Faculty, Autonomous University of Baja California, Mexicali 21000, Mexico; 6Medicine Faculty, National University of Colombia, Bogotá 111321, Colombia; leguerreroo@unal.edu.co

**Keywords:** folic acid intake, ventricular septal defects, congenital heart disease, methylation status, *AXIN1*, *MTHFR*, *TBX1*, *TBX20*

## Abstract

Background: DNA methylation is the best epigenetic mechanism for explaining the interactions between nutrients and genes involved in intrauterine growth and development programming. A possible contributor of methylation abnormalities to congenital heart disease is the folate methylation regulatory pathway; however, the mechanisms and methylation patterns of VSD-associated genes are not fully understood. Objective: To determine if maternal dietary intake of folic acid (FA) is related to the methylation status (MS) of VSD-associated genes (AXIN1, MTHFR, TBX1, and TBX20). Methods: Prospective case–control study; 48 mothers and their children were evaluated. The mothers’ dietary variables were collected through a food frequency questionnaire focusing on FA and the consumption of supplements with FA. The MS of promoters of genes was determined in the children. Results: The intake of FA supplements was significantly higher in the control mothers. In terms of maternal folic acid consumption, significant differences were found in the first trimester of pregnancy. Significant differences were observed in the MS of MTHFR and AXIN1 genes in VSD and control children. A correlation between maternal FA supplementation and MS of AXIN1 and TBX20 genes was found in control and VSD children, respectively. Conclusions: A lower MS of AXIN1 genes and a higher MS of TBX20 genes is associated with FA maternal supplementation.

## 1. Introduction

Folic acid (FA) deficiency is widespread and constitutes a significant global disease burden, which also affects women during the reproductive period [1]. The main cause of folate deficiency is poor dietary intake. Folate is an important substrate in carbon metabolism, by which carbon groups are provided for DNA methylation and DNA, RNA, proteins, and lipids synthesis [2]. These folate-dependent processes are essential during periods of rapid cell division and growth. Therefore, the requirement of folic acid during pregnancy increases markedly to satisfy the needs of embryonic and fetal growth and development.

The protective effect of periconceptional folate in preventing neural tube defects is widely recognized [3]. In recent years, the relevance of the relationship of folic acid intake with the prevention of congenital heart disease (CHD) has increased, which has led to a search for candidate genes involved in its metabolic pathway [4,5]. Four genes have recently been shown to be related to CHD: *MTHFR* (methylenetetrahydrofolate reductase), *TBX1* (T-box transcription factor 1), *TBX20* (T-Box transcription factor 20), and *AXIN1* (axis inhibitor 1) [6,7,8,9].

The MTHFR enzyme is a promising candidate because it regulates the availability of active folate by catalyzing the reduction of 5,10-methylenetetrahydrofolate to 5-methyltetrahydrofolate. Reduced MTHFR activity results in the decreased availability of 5-methyltetrahydrofolate for the re-methylation of homocysteine to methionine [10]. Several studies have shown the association of *MTHFR* variants with increased risk of isolated CHD, as well as differences in methylation patterns between promoter regions of the *MTHFR* gene in children with CHD [6].

Moreover, the *TBX1* gene encodes a T-box transcription factor implicated in DiGeorge syndrome, which affects the development of many organs, including the heart [11]. Studies have reported mutations in the *TBX1* gene in families with a history of CHD, and some mutations have been related to isolated abnormalities such as tetralogy of Fallot, interrupted aortic arch, double ventricular right ventricle, pulmonary atresia, and ventricular septal defects (VSD) [7], while the *TBX20* gene performs critical activities in the development of the heart and adult cardiac function [12]. Mutations in the coding region of *TBX20* have been associated with sporadic and familial cases of coronary heart disease, including atrial septal defects (ASD), tetralogy of Fallot, and dilated cardiomyopathy in adults [8]. Furthermore, differences in the methylation patterns between subjects with dilated heart disease and VSD have been observed in the promoter region of *TBX20* [13].

*AXIN1* is a component of the WNT signal transduction pathway and plays a role in the assembly of the b-catenin complex that regulates cell proliferation and promotes myogenesis or osteogenesis. Recent studies have found an association between the presence of allelic variants of this gene and the risk of CHD [9].

Congenital malformations are estimated to affect 7% of births [14]. VSD accounts for 20% of all CHD. The incidence is approximately 1.5 to 3.5 per 1000 newborns [15]. In Mexico, it is estimated that each year between 12 and 16 thousand children are born with CHD, and it is the second most prevalent cause of death in Mexican children under five years of age [16]. VSD is the third most common isolated defect, after patent ductus arteriosus (PDA) and inter-atrial communication (IAC) [17]. Some studies have reported a prevalence of children under 6 years of age with VSD of between 9.3 and 10.29% in Mexico [18,19].

Given the limited literature available on gene–diet interaction, and specifically on maternal folic acid intake and its relationship with the global methylation of genes associated with VSD (*MTHFR*, *TBX1*, *TBX20*, and *AXIN1*), this research aims to determine the relationship between maternal folic acid intake during pregnancy and the methylation status (MS) of genes associated with VSD.

## 2. Materials and Methods

### 2.1. Study Population

A census was conducted at 2 local hospitals: Hospital Universitario “Dr. José Eleuterio González” and Hospital Regional Materno-Infantil de Alta Especialidad, in Nuevo León, Mexico, in 2010–2012. It found 50 children with CHD. For this study, all VSD patients were recruited (*n* = 16; 32% of all CHD). The study was approved by the Ethics and Research Committee of the Facultad de Medicina, Universidad Autónoma de Nuevo León (GE19-00001), according to The Code of Ethics of the World Medical Association (Declaration of Helsinki) [20]. Informed consent was obtained from all subjects involved in the study, as established in the Regulations of the General Health Law of Mexico [21].

This prospective case–control study included all children with a positive diagnosis of VSD and their mothers. Patients with a positive family history of CHD or with a clear environmental risk factor related to the malformation were excluded. Patients with other malformations that suggest a chromosomal or monogenic etiology for CHD and subjects with additional symptoms were excluded. The controls were DNA samples previously collected from healthy children from mothers whose medical history and dietary information was known, which were stored in the DNA Bank of the Department of Genetics of the Hospital Universitario ‘‘Dr. José Eleuterio González”.

### 2.2. Determination of Maternal Dietary Intake of Folic Acid

Via direct interview, the mother completed a data collection sheet, which included the qualitative determination of the consumption of folic acid supplements before and during pregnancy (nonusers were defined as those who never took folic acid supplements alone or folic acid-containing multivitamins before conception and/or during pregnancy; users were defined as those who took folic acid supplements alone or folic acid-containing multivitamins before conception and/or during pregnancy), the presence of maternal diabetes mellitus, alcohol intake, and exposure to medications during pregnancy. The quantitative determination of folic acid derived from diet (including foods with folate and foods fortified with folic acid) was carried out using a food frequency questionnaire pertaining to the consumption of foods high in folic acid; this contained a list of 45 foods and beverages, validated for the Mexican female population by the Facultad de Salud Pública y Nutrición, Universidad Autónoma de Nuevo León, through which an estimate of the weekly intake of folic acid was derived. Subsequently, each questionnaire was evaluated using the Food Processor software (ESHA’s Food Processor^®^ Nutrition Analysis software, ESHA Research, Salem, OR, USA), which shows the weekly dietary intake of folic acid expressed in micrograms (mcg); 400 mcg/day was considered the cut-off point of FA intake [22].

### 2.3. Selection of Genetic Variants

A previous study carried out by our group (Hernández-Almaguer et al. 2019) found statistically significant differences in genetic polymorphisms of the *TBX20* and *AXIN1* genes between cases and controls with a significant increase in the risk of congenital septal heart defects in the population from northeast Mexico [9]. The *MTHFR* and *TBX1* genes, selected from the existing literature, might contribute to the presence of VSD [23,24,25,26,27,28].

### 2.4. DNA Extraction and Genotyping

DNA was extracted from blood samples (approximately 10 drops) obtained from infants during neonatal screening using the commercial Wizard Genomic DNA purification kit (Promega, Madison, WI, USA), following the manufacturer’s procedures. DNA quality and quantity were also verified by spectrophotometry (UV–Vis) with the NanoDropTM 8000 (Thermo Fisher, Wilmington, DE, USA).

### 2.5. Quantitative Methylation Analysis

Methylation analysis of specific gene promoters was performed using the EpiJET™ DNA Methylation Analysis Kits (MspI/HpaII) (Thermo Scientific™ Vilnius, Lithuania), following the manufacturer’s instructions. A DNA sample was divided into 3 tubes, each digested with a different endonuclease cut from either: (1) methylated DNA; (2) unmethylated DNA; or (3) undigested DNA. Methylated and unmethylated DNA controls were also included. We incubated samples for 1 h at 37 °C.

### 2.6. Methylation Specific PCR

Quantitative PCR (qPCR) was used to estimate the methylation level of the *MTHFR, TBX1, TBX20,* and *AXIN1* gene promoters, using Sybr Green with the following specific primers: TBX1-Fod—AATGGGCGTCTTGTCTTCGC, TBX1-Rev—GGGTCGCAGGGTCTGATTCC; TBX20-Fod—CTGTGCAGACTGTCGTCCTG, TBX20-Rev—CACTGGCCTCTATTCCCCAC; MTHFR-Fod—GGGCCTGAGCTGACAGAGAT, MTHFR-Rev—AACATGCTCCTCGGTGACAG; AXIN1-Fod—ATGTCAGCCCCTTGTTTTTGCT, and AXIN1-Rev—ATCTCGGGTAGCCGGTTTAGACT. The reaction was carried out in a final volume of 20 µL, containing 40 ng of digested DNA. QPCR was performed in a Step One Plus thermocycler (Applied biosystems^TM^ Foster City, CA, USA). The thermocycler protocol was as follows: DNA denaturation by incubation at 95 °C for 10 min, followed by 40 cycles of 95 °C for 15 s, 60 °C for 30 s, 72 °C for 30 s and a final extension at 72 °C for 2 min. The methylation of the promoters of the genes of interest was analyzed using the StepOne Real-Time PCR System (Applied biosystems^TM^ Foster City, CA, USA). Methylation was quantitatively expressed as the percentage of methylated cytosines over the total number of methylated and unmethylated cytosines.

### 2.7. Statistical Analysis

Data normality tests were performed, and descriptive statistics and frequencies were also obtained for quantitative and qualitative variables, respectively. To determine the statistical differences between the cases and controls in terms of qualitative variables, the Chi^2^ test was performed, while the Mann–Whitney U or Student’s *t*-test was used for the quantitative variables, depending on the distribution of the data. OR values were obtained to determine the association between folate intake and the risk of VSD. To determine the association between the start time of maternal folic acid consumption and the presence of VSD, Fisher’s exact test was performed. To correlate the dietary intake of maternal folic acid and the MS of VSD-associated genes, Spearman correlation was performed. Statistical analysis was performed in the SPSS v.22 statistical package (SPSS Inc.; Chicago, IL, USA). Probability values less than 0.05 (*p* < 0.05) were considered statistically significant.

## 3. Results

The case group included 16 children with VSD (56.3% male and 43.8% female) and their mothers, while the control group contained 32 binomials (mother and child). The clinical and dietary characteristics of both groups are shown in Table 1.

Older maternal age was reported in mothers of children with VSD, but no significant differences were found (25.22 ± 7.21 vs. 20.96 ± 3.03; *p* = 0.072). No significant differences in maternal dietary intake of global FA were observed between the groups (*p* = 0.600). However, a significant difference was found in the consumption of FA supplements; a higher proportion of mothers of neonates with VSD had a lower intake (*p* = 0.001; Table 1), which increased the risk of having a child with VSD (OR = 3909, 95% CI = 2.348–6.508; Table 2).

In the segmentation analysis of the start time of maternal folic acid consumption, significant differences were found in the first trimester of pregnancy (*p* = 0.0255). There were no data for the start time of folic acid supplement consumption for one mother, thus these were excluded from the analysis (Table 3).

In the analysis of the global MS, significant differences were observed between healthy children and children with VSD in terms of *MTHFR* and *AXIN1* genes (0.88 ± 1.71 vs. 3.32 ± 4.44, *p* = 0.001; 89.57 ± 42.52 vs. 58.74 ± 28.47, *p* = 0.012, respectively; Table 4). However, no correlation was found between maternal dietary intake of folic acid and the MS of any gene studied (Table 5).

Regarding maternal intake of folic acid supplements, five women with a VSD child reported no consumption of folic acid supplements (folic acid supplements alone or folic acid-containing multivitamins before conception and/or during pregnancy)—they were excluded from the analysis of the association between maternal intake of folic acid supplements and the MS of genes associated with VSD. Two genes were proven to be associated with VSD in healthy children as opposed to children with VSD. *AXIN1* displayed a lower MS in VSD children (89.57 ± 42.52 vs. 55.80 ± 29.99, *p* = 0.020; control vs. VSD, respectively), while *TBX20* displayed a higher MS: 1.08 ± 0.81 vs. 1.87 ± 1.77, *p* = 0.049 (Table 6).

## 4. Discussion

To our knowledge, this is the first published study in which the relationship between maternal dietary folic acid consumption and the MS of genes associated with VSD in children is analyzed.

Compared to the recommended intake of folic acid in reproductive-age women, a low dietary folic acid intake (<400 mcg/day) in the studied women did not relate to a risk of VSD in their children (OR = 1.571, 95% CI = 0.409–6.040). This is probably because the maternal dietary folic acid intake per day was very similar in both groups (controls: 372.02 ± 241.14 vs. cases: 346.72 ± 152.10, *p* = 0.600). However, no maternal consumption of supplements with folic acid was a risk factor for VSD in children (OR = 3.909, 95% CI = 2.348–6.508). This opposes what was reported by Mao et al. (2017), who determined that there was an increased risk for those who in the lowest quartile of dietary folate intake during pregnancy compared with the middle quartiles (25th to 75th) (OR: 1.63, 95% CI: 1.01 ± 2.62) [29]. In addition, compared to nonusers, folic acid supplement users did not exhibit a significantly reduced risk of CHDs (OR: 0.81, 95% CI: 0.51–1.29).

Although no differences were observed in maternal folic acid dietary intake between cases and controls, the intake of folic acid supplements in mothers, regardless of the time of commencement or type of supplement (folic acid alone or in a multivitamin), seems to be related to differences in methylation percentage in two of the four genes studied (*AXIN1: p* = 0.020 and *TBX20: p* = 0.049).

As regards gene methylation, *MTHFR* has been reported to be associated with a risk of CHD in Down syndrome children [30]. In addition, with an increase in promoter methylation level in *MTHFR*, the MTHFR protein’s activity is reduced, thereby increasing the risk of various diseases. An increase *MTHFR* promoter methylation was also seen in DNA isolated from cancer patients, patients with cardiovascular or renal disorders, and placental DNA from women with pre-eclampsia [31,32]. In our study, a higher methylation percentage was observed in children with VSD compared to healthy children, with significant differences (3.32 ± 4.44 vs. 0.88 ±1.71, *p* = 0.001).

Previous studies have demonstrated that *TBX20* is an important transcription factor with a highly conserved DNA-binding region (T-box), and it plays an essential role in the development of CHD in humans. During development, the *TBX20* gene is expressed in the atrioventricular channel, the outflow tract, and the developing right ventricle and valves [33,34]. Recently, hypomethylation of the *TBX20* promoter region was observed in the tetralogy of Fallot patients [35], but no studies have been undertaken on VSD patients. We found a low methylation percentage in both groups (cases and controls), but no statistically significant differences between them were observed (cases: 2.12 ± 2.16 vs. controls: 1.08 ± 0.81, *p* = 0.090).

*AXIN1* encodes a cytoplasmic protein that inhibits the Wnt signaling pathway [36,37]. An association has been reported between the hypermethylation of the *AXIN1* promoter and caudal duplication anomalies. Oates et al. (2006) analyzed methylation in the promoter region of the *AXIN1* gene in monozygotic twins, finding a significantly more methylated promoter region in the twin with caudal duplication than in the unaffected twin [14]. Until now, no studies analyzing the association between *AXIN1* gene methylation and the presence of VSD have been reported. We found a significantly lower methylation percentage in children with VSD than in healthy children (58.74 ± 28.47 vs. 89.57 ± 42.52, *p* = 0.012). In addition, similar methylation percentage results were derived from the analysis of maternal folic acid supplement consumption (control: 89.57 ± 42.52 vs. VSD: 55.80 ± 29.99, *p* = 0.020), which seems to be related to the start time of maternal folic acid supplement consumption (*p* = 0.025).

The ventricular septum lengthens via the apposition process, meaning that cells are added to the septum at its base [38]. This process starts in the fourth week of development in humans (CS12) [39]. When apposition is abnormal, a hole or multiple holes form in the ventricular septum, referred to as muscular ventricular septal defects. Different studies have shown that periconceptional consumption of folic acid-containing multivitamins results in a significant reduction in the prevalence of CHD [40,41,42,43,44,45]. Besides this, according to Czeizel et al. (2015), there was a significant reduction in the prevalence of VSD (OR 0.57, 95% CI 0.45–0.73) in infants born to mothers who had taken high doses of folic acid during the critical period of CHD development [46]. Moreover, low folate status is associated with reductions in global DNA methylation, a covalent modification of genomic DNA that affects gene expression [47]. The correct expression of *AXIN1* (an important negative regulator of the canonical Wnt signal transduction machinery, which is a rate-limiting factor for b-catenin destruction complex assembly) plays a crucial role during embryonic development [48,49]. The start time of folic acid supplement consumption and *AXIN1* expression could thus together play a crucial role during cardiogenesis, including cardiac septation. In our study, there was a lower proportion of maternal folic acid supplement consumption in the first trimester (74.2% vs. 54.5%, control vs. VSD, respectively), a critical period of ventricular septal formation [50].

In conclusion, although no significant differences were observed, there was a trend towards higher maternal age in children with VSD. Similarly, no differences were observed in dietary intake of folic acid; however, the consumption of FA supplements and the MS of VSD-associated genes were different between cases and controls, and FA maternal supplementation was identified as a risk factor of VSD, correlated with the MS of *AXIN1* and *TBX20* genes. Future studies should be undertaken to investigate the association of maternal FA intake (dietary and supplementary) and DNA methylation patterns in children with a risk of CHDs.

## Figures and Tables

**Table 1 nutrients-13-02071-t001:** Contrast of clinical variables and maternal folic acid intake in children.

	Presence of VSD		
	Healthy *n* = 32	VSD *n* = 16	*p*
**Gender ^1^**	***n* (%)**	***n* (%)**	
**Male**	13 (40.6)	9 (56.3)	0.306
**Female**	19 (59.4)	7 (43.8)	
**Maternal age, mean ± SD**	20.96 ± 3.03	25.22 ± 7.21	0.072
**Presence of Maternal Diabetes Mellitus ^1^**			
**Yes**	2 (6.3)	3 (18.8)	0.181
**No**	30 (93.8)	13 (81.3)	
**Family Background ^1^**			
**Heart disease**	7 (21.9)	5 (31.3)	0.845
**Diabetes mellitus II**	6 (18.8)	3 (18.8)	
**Cancer**	0 (0)	1 (6.3)	
**Thyroid disease**	1 (3.1)	0 (0)	
**Rheumatologic disease**	1 (3.1)	0 (0)	
**No hereditary family history**	17 (53.1)	7 (43.8)	
**Exposure to Medications During Pregnancy ^1^**			
**No**	23 (71.9)	11 (68.8)	0.822
**Yes**	9 (28.1)	5 (31.3)	
**Hypoglycemic drugs**	0 (0)	1 (20)	0.699
**Antipyretics**	2 (22.2)	1 (20)	
**Proton-pump inhibitor**	1 (11.1)	1 (20)	
**Antibiotics**	2 (22.2)	1 (20)	
**Systemic action antifungals**	1 (11.1)	1 (20)	
**β2 adrenergic agonists**	1 (11.1)	0 (0)	
**Antimuscarinics**	2 (22.2)	0 (0)	
**Occasional Alcoholism ^1^**			
**Yes**	5 (15.6)	0 (0)	0.095
**No**	27 (84.4))	16 (100)	
**Folic Acid Supplementation ^1^**			
**Yes**	32 (100)	11 (68.8)	0.001 *
**No**	0 (0)	5 (31.3)	
**Start of Consumption of Folic Acid Supplements ^1^**			
**Before Pregnancy**	1 (3.2)	2 (18.2)	0.309
**First Trimester**	23 (74.2)	6 (54.5)	
**Second Trimester**	6 (19.4)	3 (27.3)	
**Third Trimester**	1 (3.2)	0 (0)	
**Folic Acid Supplement Type ^1,3^**			
**Folic Acid Alone**	20 (62.5)	7 (77.8)	0.372
**Multivitamin**	6 (18.8)	2 (22.2)	
**Both**	6 (18.8)	0 (0)	
**Folic Acid Maternal Dietary Intake (mcg) ^2^**			
**Weekly, mean ± SD**	2604.11 ± 1689.81	2427.07 ± 1064.72	0.600
**Per day, mean ± SD**	372.02 ± 241.14	346.72 ± 152.10	

^1^ Chi^2^. ^2^ Mann–Whitney U. ^3^ Two women with a ventricular septal defects (VSD) child did not report the type of supplement consumed. * Statistically significant difference (*p* < 0.05). SD: standard deviation.

**Table 2 nutrients-13-02071-t002:** Associations between folate intake and the risk of VSD.

	Presence of VSD		OR	95% CI
	Healthy *n* = 32	VSD *n* = 16		
**Folic Acid Maternal Dietary Intake**	***n* (%)**	***n* (%)**		
<400 mcg/day	21 (65.6)	12 (75)	1.571	0.409–6.040
>400 mcg/day	11 (34.4)	4 (25)		
**Folic Acid Supplementation**				
No	0 (0)	5 (31.3)	3.909	2.348–6.508
Yes	32 (100)	11 (68.8)		

**Table 3 nutrients-13-02071-t003:** Associations between the start time of maternal folic acid supplement consumption and the presence of VSD.

Star Time of Maternal Folic Acid Supplement Consumption	Folic Acid Supplementation		Marginal Row Totals	*p*
	With Supplementation	Without Supplementation		
**Before Pregnancy**	***n* (%)**	***n* (%)**		
Healthy	1	30	31	0.264
VSD	2	14	16	
Marginal column totals	3	44	47	
**First Trimester**				
Healthy	23	8	31	0.025 *
VSD	6	10	16	
Marginal column totals	29	18	47	
**Second Trimester**				
Healthy	6	25	31	1
VSD	3	13	16	
Marginal column totals	9	38	47	
**Third Trimester**				
Healthy	1	30	31	1
VSD	0	16	16	
Marginal column totals	1	46	47	

* Statistically significant difference (*p* < 0.05).

**Table 4 nutrients-13-02071-t004:** VSD associated gene MS in all children.

Methylation Percentage	Presence of VSD		
	Healthy *n* = 32	VSD *n* = 16	*p* ^1^
*AXIN1*, mean ± SD	89.57 ± 42.52	58.74 ± 28.47	0.012 *
*MTHFR*, mean ± SD	0.88 ± 1.71	3.32 ± 4.44	0.001 *
*TBX1*, mean ± SD	1.37 ± 3.27	1.63 ± 1.88	0.149
*TBX20*, mean ± SD	1.08 ± 0.81	2.12 ±2.16	0.090

^1^ Mann–Whitney U. * Statistically significant difference (*p* < 0.05).

**Table 5 nutrients-13-02071-t005:** Correlation of maternal folic acid dietary intake with the methylation percentage of genes associated with VSD in children.

VSD Associated Genes	Methylation Percentage Mean ± SD	Maternal Folic Acid Intake Per Day (mg) Mean ± SD	Rho *	*p*
*AXIN1*	79.29 ± 40.83	363.59 ± 214.39	0.010	0.944
*MTHFR*	1.69 ± 3.09		0.015	0.918
*TBX1*	1.46 ± 2.86		−0.016	0.913
*TBX20*	1.42 ± 1.47		−0.161	0.273

*n* = 48. * Spearman correlation.

**Table 6 nutrients-13-02071-t006:** Association of maternal intake of folic acid supplements with the methylation percentage of genes associated with VSD in children.

Mothers with Folic Acid Supplementation **	Presence of VSD	Methylation Percentage Mean ± SD	*p* ^1^	95% CI	
				Lower	Higher
*AXIN1*	Healthy (*n* = 32)	89.57 ± 42.52	0.020 *	5.656	61.881
	VSD (*n* = 11)	55.80 ± 29.99			
*MTHFR*	Healthy (*n* = 32)	0.88 ± 1.71	0.091	−6.564	0.571
	VSD (*n* = 11)	3.88 ± 5.27			
*TBX1*	Healthy (*n* = 32)	1.37 ± 3.27	0.843	−2.328	1.910
	VSD (*n* = 11)	1.58 ± 1.96			
*TBX20*	Healthy (*n* = 32)	1.08 ± 0.81	0.049 *	−1.589	−0.002
	VSD (*n* = 11)	1.87 ± 1.77			

^1^ Student’s *t*-test. * Statistically significant difference (*p* < 0.05). ** Five of the mothers of children with VSD did not take folic acid supplements.

## Data Availability

Not applicable.

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
