# Peer review of "Maternal Folic Acid Intake and Methylation Status of Genes Associated with Ventricular Septal Defects in Children: Case–Control Study"

_nutrients, 2021, doi:10.3390/nu13062071_

Round 1
Reviewer 1 Report
On the section: Determination of maternal dietary intake of folic acid, please correct the next minor methodological issues: To add version of The Food Processor software . It is recommended to change the term quiz by food frequency questionnaire. Also, to correct unit of measurement of the folic acid intake (it says milligrams and must be: micrograms). Please, to specify the reference of the cut-off point for folic acid intake (<400 mcg and > 400 mcg).
On the tables: Check all bold items that are unnecessary. On table 3: correct 0.0255 (only three decimal places).
Author Response
1- To add version of The Food Processor software.
Response: Done Line 119
2- It is recommended to change the term quiz by food frequency questionnaire.
Response: Done Line 114
3- Also, to correct unit of measurement of the folic acid intake (it says milligrams and must be: micrograms).
Response: Done Line 121
4- Please, to specify the reference of the cut-off point for folic acid intake (<400 mcg and > 400 mcg).
Response: Done Line 219
5- On the tables: Check all bold items that are unnecessary. On table 3: correct 0.0255 (only three decimal places).
Response: Done
Reviewer 2 Report
This is certainly an interesting manuscript that brings to light some novel findings of the importance of folate in the correct development of the embryo, specifically in a structure such as the heart whose malformations cause serious consequences. I think this is a well-conducted study that adds some new information to the role of maternal folic acid in ventricular septal defects.
Although the content of the manuscript is very interesting, some proposals such as those detailed below can improve your exposure:
Title: It is better if the abbreviation “VSD” doesn´t appear in the title.
Line 36: This is the first time that “folic acid” has been written but not its abbreviation. The authors use this term several times up to line 180, where “FA” appears.
Although the abbreviation “FA” appears in the abstract, it must be placed the first time it appears in the main text (line 36) and used in the rest of the text.
Lines 73-74: “ventricular septal defect (VSD)” is written the same on lines 62-63. From these lines, it can be used the abbreviation.
Sometimes authors wrote “ventricular septal defect” and other “ventricular septal defects” (with “s” at the end of “defect”). Please check this.
Line 120: What is the data of “ESHA’s Food Processor® Nutrition Analysis Software”?
Line 124: Do not write a comma between the word "the" and the word "TBX20".
Line 132: Add city and country of “Promega”.
Line 137: What is the data of “EpiJet 136 DNA Methylation Analysis (MspI / HpaII) kit”?
Line 151: Add city and country of “Applied Biosystem”.
Line 155: What is the data of “Software Step One v2.2.2”?
Line 262: Please, check this sentence.
Line 281-282: The authors write “…the first trimester…a critical period of ventricular septal formation”. Please, add a reference.
Line 287: In the abstract and on this line only, the authors use the abbreviation “MS” for “methylation status". I propose not to use the abbreviation for this term or to add "methylation status (MS)" on line 192.
Author Response
Title: It is better if the abbreviation “VSD” doesn´t appear in the title.
Response: Done
Line 36: This is the first time that “folic acid” has been written but not its abbreviation. The authors use this term several times up to line 180, where “FA” appears.
Although the abbreviation “FA” appears in the abstract, it must be placed the first time it appears in the main text (line 36) and used in the rest of the text.
Response: Done
Lines 73-74: “ventricular septal defect (VSD)” is written the same on lines 62-63. From these lines, it can be used the abbreviation.
Response: Done
Sometimes authors wrote “ventricular septal defect” and other “ventricular septal defects” (with “s” at the end of “defect”). Please check this.
Response: Done
Line 120: What is the data of “ESHA’s Food Processor® Nutrition Analysis Software”?
Response: ESHA’s Food Processor® Nutrition Analysis software, ESHA Research, Salem, Oregon, USA
Line 124: Do not write a comma between the word "the" and the word "TBX20".
Response: Done
Line 132: Add city and country of “Promega”.
Response: Done (Promega Madison, WI, USA)
Line 137: What is the data of “EpiJet 136 DNA Methylation Analysis (MspI / HpaII) kit”?
Response: (Thermo Scientific™ Vilnius, Lithuania)
Line 151: Add city and country of “Applied Biosystem”.
Response: (Applied biosystemsTM Foster City, CA, USA)
Line 155: What is the data of “Software Step One v2.2.2”?
Response: (Applied biosystemsTM Foster City, CA, USA)
Line 262: Please, check this sentence.
Response: Done
Line 281-282: The authors write “…the first trimester…a critical period of ventricular septal formation”. Please, add a reference.
Response: Done
Line 287: In the abstract and on this line only, the authors use the abbreviation “MS” for “methylation status". I propose not to use the abbreviation for this term or to add "methylation status (MS)" on line 192.
Response: Done
Reviewer 3 Report
This interesting, original study addresses to determine if maternal dietary intake of folic acid is related with methylation status of ventricular septal defects (VSD) associated genes (AXIN1, MTHFR, TBX1 22 and TBX20).
In my opinion, the most important limitation of the study is the number of patients included (only 16 vs. 32 controls). Authors should be mention this specific limitation of their study in the "Discussion" section. In the “Abstract”, authors should also clarify that the 48 mothers and their children are divided in this two groups (16 VSD patients vs. 32 healthy controls). Furthermore, they should consider asking some specific questions that could be addressed in future research.
The manuscript should be 'spell checked' and 'grammatically checked'. I have found several examples that the manuscript has not been reviewed by a native English speaker (e.g. in line 20, line 123 –reference should be substitute by a number–, line 124, line 159, line 215, line 286, and the legend of Table 1). In addition, authors use (two times) “on the other hand” to refer to something additional. In English, on the other hand is used to introduce a statement that contrasts with a previous statement or presents a different point of view. Thus, on the other hand is often used after a statement introduced with on the one hand. Therefore, authors should replace “on the other hand” by “moreover”, “furthermore”, “in addition”, “also”, “additionally”, etc. (e.g. in line 58 and line 273).
Please use a decimal point and not a comma to indicate the decimal place (e.g. in line 218 and line 222).
As I am not a native English speaker either, I request authors to submit the manuscript to a scientific proofreading service in accordance with the standards of the journal.
Author Response
1-The most important limitation of the study is the number of patients included (only 16 vs. 32 controls). Authors should be mention this specific limitation of their study in the "Discussion" section.
Response: Done
2-In the “Abstract”, authors should also clarify that the 48 mothers and their children are divided in this two groups (16 VSD patients vs. 32 healthy controls).
Response: The number of characters is limited by the journal itself, for that reason we not explained in the abstract.
3-Furthermore, they should consider asking some specific questions that could be addressed in future research.
Response: Done
4-The manuscript should be 'spell checked' and 'grammatically checked'. I have found several examples that the manuscript has not been reviewed by a native English speaker (e.g. in line 20, line 123 –reference should be substitute by a number–, line 124, line 159, line 215, line 286, and the legend of Table 1). In addition, authors use (two times) “on the other hand” to refer to something additional. In English, on the other hand is used to introduce a statement that contrasts with a previous statement or presents a different point of view. Thus, on the other hand is often used after a statement introduced with on the one hand. Therefore, authors should replace “on the other hand” by “moreover”, “furthermore”, “in addition”, “also”, “additionally”, etc. (e.g. in line 58 and line 273).
Response: We are going to submit the manuscript to the scientific proofreading service of the journal own.
5-Please use a decimal point and not a comma to indicate the decimal place (e.g. in line 218 and line 222).
Response: Done
6-As I am not a native English speaker either, I request authors to submit the manuscript to a scientific proofreading service in accordance with the standards of the journal.
Response: We are going to submit the manuscript to the scientific proofreading service of the journal own.